# Design and Characteristic Analysis of a MEMS Piezo-Driven Recirculating Inkjet Printhead Using Lumped Element Modeling

**DOI:** 10.3390/mi10110757

**Published:** 2019-11-06

**Authors:** Muhammad Ali Shah, Duck-Gyu Lee, Shin Hur

**Affiliations:** 1Nano-Mechatronics, University of Science and Technology, Daejeon 34113, Korea; ali@kimm.re.kr; 2Korea Institute of Machinery and Materials, 156 Gajeongbuk-Ro, Yuseong-Gu, Daejeon 34103, Korea; educk9@kimm.re.kr

**Keywords:** recirculating, LEM, piezoelectric, inkjet printhead

## Abstract

The recirculation of ink in an inkjet printhead system keeps the ink temperature and viscosity constant, and leads to the development of a high-performance device. Herein, we propose a recirculating piezo-driven micro-electro-mechanical system (MEMS)-based inkjet printhead that has a pressure chamber, a nozzle, and double restrictors. The design and characteristic analysis are performed using a two-port lumped element model (LEM) to investigate the effect of design parameters on the system responses. Using LEM, the jetting pressure at the pressure chamber, velocity at the nozzle inlet, meniscus pressure, and Helmholtz resonance frequency are predicted and the comparative analysis of the jetting pressure and velocity between LEM and the finite element method (FEM) simulation is conducted to validate our proposed LEM method. Furthermore, the effect of a change in major design parameters on the jetting pressure, velocity, and Helmholtz resonance frequency is analyzed. On the basis of this analysis, the optimized device dimensions are finalized. From our analysis, it is also concluded that the restrictor is more sensitive than the pressure chamber in terms of their variations in depth. As the cross-talk effect can occur due to an array of hundreds or thousands of nozzles, we investigated the effect of a single activated nozzle on the non-activated neighboring nozzles, as well as the effect of multi-activated nozzles on a single central nozzle using our proposed LEM.

## 1. Introduction

Recent trends in the development of dyeing process technology and its application in new facilities show the improvement of environmentally friendly and economical characteristics. Therefore, digital textile printing (DTP) technology, which can reduce the consumption of water and raw materials and simplify the process, is becoming more important. By definition, digital printing digitizes the entire process from design to printing such that printing can be done with inkjet printers that are filmless and untreated [1]. The inkjet prints used in digital printing can be divided into the drop-on-demand (DOD) type and the continuous type depending on the head characteristics of the print. There are piezoelectric and thermal inkjet printheads in the DOD system. The piezoelectric inkjet printhead (PIP) can handle a larger variety of inks compared to thermal printheads and is mainly used in industrial applications [1,2,3,4].

PIP involves a moving membrane, a restrictor, a pressure chamber, and a nozzle. In PIP, an electrically deformed piezoelectric membrane generates a pressure inside a pressure chamber. Due to the pressure, ink is injected into the nozzle. The interaction of fluid with the microchannels makes the system complex and highly dynamic. This interaction needs to be modeled and well-characterized to investigate the behavior of a printhead. The complexity increases in the presence of arrays of hundreds or thousands of nozzles. To treat these complex systems, lumped element modeling (LEM) can be used.

Several researchers conducted research on design and analysis. Prasad et al. [5] have studied two-port LEM for a piezoelectric transducer. In their intensive element modeling, the individual components of a piezoelectric transducer were modeled as elements of an equivalent electrical circuit using conjugate power variables, and the synthesis of the two-port model is performed by determining the transverse static strain field as a function of pressure and voltage loads. In addition, methods for estimating model parameters were discussed. Berger and Recktenwald [6] investigated the Helmholtz and driving resonant frequencies of a PIP using LEM and presented an improved model using transmission lines when predicting the parasitic frequencies. Kim et al. [7] presented PIP that has a nozzle at one end and the other end connected to the reservoir through a narrow rectangular channel called a restrictor. The pressure and the velocity profiles were acquired through analysis of numerical and lumped models. Wang et al. [8] performed LEM for the squeeze mode PIP. The device is actuated by squeezing the piezoelectric membrane. Furthermore, the double trapezoidal pulse waveform was presented to suppress the residual oscillations of the meniscus occurred after formation of the main droplet.

The LEM analyzed designs presented in references [6,7,8], as well as some other designs with improved dots-per-inch (DPI) and waveforms [9,10], and in miscellaneous studies [11,12,13], are non-recirculating. The problem with non-recirculating PIPs is that heat can be produced during the operation and damage the nearby structure, and thereby degrade the overall performance. In addition, the nozzle can dry up and become clogged, producing a jetting failure during device operation.

The problem of heat dissipation can be solved using a recirculating piezoelectric inkjet printhead (RPIP) design, in which the fluid recirculates and dissipates the heat to a better location outside the pressure chamber. Additionally, clogging of nozzles due to air and debris in the ink can possibly be reduced in the RPIP designs. Although there are patents on RPIPs, the intensive structural analysis using LEM is missing in the literature to the best of the authors’ knowledge. However, such an analysis is needed in order to investigate the performance of these designs. Arango et al. [14] presented a full recirculating inkjet printhead system with dynamic analysis. Their model contains a full system with three tanks, in which they have analyzed the parameter changing effect outside of the printhead. The intensive analysis of the inside inkjet printhead is missing in their presented paper too.

This paper presents lumped element modeling and numerical analysis of a double-restrictor RPIP (D-RPIP) design through which its performance is investigated. Specifically, here we use double restrictors at the inlet and the outlet instead of a single restrictor. Due to such use of double restrictors, the problem of ink stagnation can be solved, and the hindrance will be discharged completely from the channel [15]. How changing the major design dimensions influences the Helmholtz resonant frequency, the pressure, and the velocity was investigated, and the comparative analysis was performed. New findings and optimized device dimensions in the case of a recirculating piezo-driven inkjet printhead are presented.

The paper is organized in the following way. The second section presents the proposed design with its parameters being identified and the equivalent circuit and its parameters, as well as the derivation of the first order differential equations of the state space model using nodal analysis. The third section that contains the results and the discussion is divided into five subsections. The first subsection covers the system responses and their comparison with finite element method (FEM) simulations to show the accuracy of our proposed lumped modeling. The second subsection includes the calculation and simulation of the Helmholtz resonance frequency, as well as analysis of the effect on it of changing major design parameters. The effect of changing design parameters on the pressure and velocity and comparative analysis between the pressure chamber and restrictor variation effects are presented in the third subsection. The fourth subsection presents the effect of changing the voltage waveform on the chamber pressure. Using LEM, the hydro-acoustic cross-talk of multi-recirculating printheads connected to each other through a common flow channel in the form of an array is presented in the fifth subsection.

## 2. Materials and Methods

### 2.1. Proposed Design

The three-dimensional view of a D-RPIP design is shown in Figure 1, in which the pressure chamber behind the nozzle was connected to the reservoir using double restrictors. The top view of the proposed D-RPIP design is shown in Figure 2a, and the cross-section view of the central position of the device showing the piezo-membrane, the pressure chamber, and the nozzle is given in Figure 2b. In the same figure, the piezoelectric material Lead Zirconate Titanate (PZT), sandwiched between the upper and lower electrodes, is shown. By applying a voltage to the PZT plate, the membrane deforms, generating a pressure wave in the pressure chamber that travels toward the nozzle, which speeds up the jetting velocity, forming a droplet or a series of droplets at the nozzle exit. The fluid flow resistances in these microchannels and the mechanical loss of the piezo-membrane are presented in Figure 2c. The dimensions of the restrictor, pressure chamber, and nozzle are given in Table 1, and the piezo-membrane materials and structural dimensions are presented in Table 2.

### 2.2. Lumped Element Model

Electrical, mechanical, and fluidic components are three different energy domains involved in the inkjet printheads. The behavior of the fluid in the lumped modeling can be described using three parameters: capacitance (C), inductance (L), and resistance (R), related to compressibility, mass, and resistance of fluid, respectively. In our proposed model, the compressibility of the small volume of the fluid inside the pressure chamber was represented using acoustic compliance, noted as $C_{p}$, which charged and discharged akin to a capacitor. This charging and discharging were due to the in and out fluid flows. At the nozzle exit, an interface between the fluid and air is formed that creates a meniscus due to surface tension. The fluid stored in the meniscus was represented by a capacitive term $C_{ms}$ and was given by the following equation [7]:(1)$$C_{ms}=\frac{V_{men}}{P_{men}}=\frac{1/2\times \left(4\pi r_{n}{}^{3}\right)/3}{2\sigma /r_{n}}=\frac{\pi r_{n}{}^{4}}{3\sigma }$$
where $V_{men}$ is the half sphere volume of the fluid with the nozzle radius $r_{n}$, $P_{men}$ is the Laplace pressure, and σ is the coefficient of surface tension. The fluid volume displaced was a half-sphere and that with such a volume displacement, the capillary pressure was defined by a spherical dome of height equal to the nozzle radius. For small fluid displacements, the meniscus generates a kind of linear stiffness for larger displacements where the capillary pressure does not change while the volume displaced increases accordingly.

The mechanical compliance, inductance, and loss of the piezo-membrane were represented using $C_{m}$, $L_{m}$, and $R_{m}$ respectively. The inductive and resistive terms of the fluid inside the restrictors, pressure chamber, and nozzle were denoted as $L_{r}$ and $R_{r}$, $L_{p}$ and $R_{p}$, and $L_{n}$ and $R_{n}$, respectively where the subscripts “*r*”, “*p*”, and “*n*” were used for the restrictor, pressure chamber, and nozzle. Generally, the fluidic inductance can be written as follows [16]:(2)$$L=\frac{\rho l}{A}$$
where l is the length of the channel, A is the cross-sectional area of the channel, and ρ is the density of the fluid.

According to Ingard [17], the experimentally verified end correction for a slot with a flanged end is given by: (3)$$l_{e}=0.48\sqrt{A}\left(1-1.25\xi \right)$$
such that the corrected length is $l_{c}=l+l_{e}$. For a rectangular aperture with a large flanged end, the term “ξ” goes to zero [17]. Therefore, the corrected inertance for the mass effect for the restrictor used in our proposed design became:(4)$$L_{r}=\frac{\rho }{A_{r}}\left(l_{r}+0.48\sqrt{A_{r}}\right)$$
The mathematical equations and values of the fluidic capacitances, inductances, and resistances used in the LEM are presented in Table 3. Due to the complex geometry, the flow resistances of the pressure chamber, the restrictor, and the nozzle used in LEM were obtained from the FEM simulations. The properties of ink used in LEM and FEM simulations are presented in Table 4, where $C_{eff}$ is the effective speed of sound, which depended on the original speed of sound (c) of the ink, initial volume ($V_{0}$), displaced volume $\left(\Delta V_{0}\right)$, and applied pressure. Due to the conical shape of the nozzle, the averaged value of the radius was used for finding the fluidic mass ($L_{n}$) of the nozzle in the LEM.

The equivalent circuit model of the proposed D-RPIP design, in which all the fluid flow in the restrictors, pressure chamber, and nozzle made a parallel configuration, is shown in Figure 3. As the ink volume at the restrictor and the nozzle was much smaller than in the pressure chamber, the capacitive term was ignored, and only the inductive and resistive terms were used for the flows in all of the restrictors and the nozzle.

The differential pressure *P* across the fluid mass *L,* and volume flow rate *Q* across the acoustic compliance *C*, were analogous to voltage *V* and current *I*, respectively, in the sense of an electric-fluid analogy, which could be expressed as: (5)$$P=L\frac{dQ}{dt}$$
(6)$$Q=C\frac{dP}{dt}$$
To find the differential terms, Equations (5) and (6) could be rewritten in the form of Equations (7) and (8), respectively,
(7)$$\frac{dQ}{dt}=\frac{P}{L}$$
(8)$$\frac{dP}{dt}=\frac{Q}{C}$$
Eleven state variables were defined, where the variables represented by ${\vec{x}}_{i}\left(t\right)$ were assigned to the volume flow rates, $x_{i}\left(t\right)$ for the pressure differences, and the subscripts denoted the position, as it is shown in Figure 3. The first-order differential equations of these state variables were obtained by solving the equivalent circuit using nodal analysis [18] and Equations (7) and (8):(9)$$\frac{dx_{1}}{dt}=\frac{1}{C_{e}}\left(\left(\frac{V-x_{1}}{R_{e}}\right)-\vec{x_{3}}\right)$$
(10)$$\frac{dx_{2}}{dt}=\frac{1}{C_{m}}\left(\vec{x_{3}}\right)$$
(11)$$\frac{d\vec{x_{3}}}{dt}=\frac{1}{L_{m}}\left(\varphi x_{1}-x_{2}-\vec{x_{3}}R_{m}-x_{7}\right)$$
(12)$$\frac{d\vec{x_{4}}}{dt}=\frac{1}{L_{ri1}}\left(x_{7}-\vec{x_{4}}R_{ri1}\right)$$
(13)$$\frac{d\vec{x_{5}}}{dt}=\frac{1}{L_{ri2}}\left(x_{7}-\vec{x_{5}}R_{ri2}\right)$$
(14)$$\frac{d\vec{x_{6}}}{dt}=\frac{1}{L_{p}}\left(x_{7}-\vec{x_{6}}R_{p}\right)$$
(15)$$\frac{dx_{7}}{dt}=\frac{1}{C_{p}}\left(\vec{x_{3}}-\vec{x_{4}}-\vec{x_{5}}-\vec{x_{6}}-\vec{x_{8}}-\vec{x_{9}}-\vec{x_{10}}\right)$$
(16)$$\frac{d\vec{x_{8}}}{dt}=\frac{1}{L_{ro1}}\left(x_{7}-\vec{x_{8}}R_{ro1}\right)$$
(17)$$\frac{d\vec{x_{9}}}{dt}=\frac{1}{L_{ro2}}\left(x_{7}-\vec{x_{9}}R_{ro2}\right)$$
(18)$$\frac{d\vec{x_{10}}}{dt}=\frac{1}{L_{n}}\left(x_{7}-\vec{x_{10}}R_{n}-x_{11}\right)$$
(19)$$\frac{dx_{11}}{dt}=\frac{1}{C_{ms}}\left(\vec{x_{10}}\right)$$
In the previous equations, V is the applied voltage, and $C_{e}$ and $R_{e}$ are the electric capacitance and resistance, respectively. $R_{ri1}$, $R_{ri2}$ and $R_{ro1}$, $R_{ro2}$ are the flow resistances of two parallel inlet restrictors and two parallel outlet restrictors in the D-RPIP design, respectively. The subscripts “*ri1*”, “*ri2*”, “*ro1*”, and “*ro2*” represent the first inlet restrictor, the second inlet restrictor, the first outlet restrictor, and the second outlet restrictor, respectively. In our LEM analysis, the two-port model represented a coupling between the electrical and fluidic domains. These two energy domains made a transformer with a turns ratio of φ, known as the electroacoustic transduction coefficient [19] that can be represented using: (20)$$\varphi =\frac{d_{a}}{C_{m}}$$
where the effective acoustic piezoelectric coefficient $d_{a}=\Delta V_{0}/V$ for which P=0 and the acoustic compliance of the PZT plate $C_{m}=\Delta V_{0}/P$ for which V=0. The values of flow resistances of the restrictor, the pressure chamber, and the nozzle, applied to the LEM, were obtained from FEM simulations using the below equation, which is equivalent to Ohm’s law ($R=\frac{V}{I}$) in the electrical domain:(21)$$R_{f}=\frac{\Delta P}{Q}$$
where $R_{f}$ is the fluid resistance, ΔP is the pressure difference, and Q is the volumetric flow rate.

By solving the state space model, the eleven state variables were expressed as the state equations of a linear time-varying system:(22)$$\dot{x}\left(t\right)=Ax\left(t\right)+Bu\left(t\right)$$
where
$A=\left[\begin{array}{ccccccccccc} -1/C_{e}R_{e} & 0 & -1/C_{e} & 0 & 0 & 0 & 0 & 0 & 0 & 0 & 0\\ 0 & 0 & -1/C_{m} & 0 & 0 & 0 & 0 & 0 & 0 & 0 & 0\\ \varphi /L_{m} & -1/L_{m} & -R_{m}/L_{m} & 0 & 0 & 0 & -1/L_{m} & 0 & 0 & 0 & 0\\ 0 & 0 & 0 & -R_{ri1}/L_{ri1} & 0 & 0 & 1/L_{ri1} & 0 & 0 & 0 & 0\\ 0 & 0 & 0 & 0 & -R_{ri2}/L_{ri2} & 0 & 1/L_{ri2} & 0 & 0 & 0 & 0\\ 0 & 0 & 0 & 0 & 0 & -R_{p}/L_{p} & 1/L_{p} & 0 & 0 & 0 & 0\\ 0 & 0 & 1/C_{p} & -1/C_{p} & -1/C_{p} & -1/C_{p} & 0 & -1/C_{p} & -1/C_{p} & -1/C_{p} & 0\\ 0 & 0 & 0 & 0 & 0 & 0 & 1/L_{ro1} & -R_{ro1}/L_{ro1} & 0 & 0 & 0\\ 0 & 0 & 0 & 0 & 0 & 0 & 1/L_{ro2} & 0 & -R_{ro2}/L_{ro2} & 0 & 0\\ 0 & 0 & 0 & 0 & 0 & 0 & 1/L_{n} & 0 & 0 & -R_{n}/L_{n} & -1/L_{n}\\ 0 & 0 & 0 & 0 & 0 & 0 & 0 & 0 & 0 & 1/L_{nz} & 0 \end{array}\right]$, $B=\left[\begin{array}{c} 1/C_{e}R_{e}\\ 0\\ 0\\ 0\\ 0\\ 0\\ 0\\ 0\\ 0\\ 0\\ 0 \end{array}\right]$, and u(t) is the input as the applied waveform.

The output equation could be written as:(23)y(t)=Cx(t)
where *C* is the output matrix for finding the pressure and volume flow rate at the pressure chamber, restrictor, and nozzle.

### 2.3. FEM Simulations

The three-dimensional model was simulated using numerical simulation software, COMSOL (4.3b, COMSOL Inc, Stockholm, Sweden). Physics of fluid-structure interaction was used by considering the fluid as laminar and incompressible. As the design was symmetric, therefore, under laminar flow, symmetry was used as a boundary condition by considering the half of the three-dimensional model to reduce the simulation time. As the pressure chamber was pressurized by deforming the piezo-membrane, a moving mesh was used by defining a mesh displacement on the wall at the top of the pressure chamber. The model was meshed by using tetrahedral, triangular, and swept meshes. Structured mesh was used for the fluid in the restrictors, nozzle, and pressure chamber. The meshed model is shown in Figure 4.

The fluid flow in the micro-channels was described using the continuity and Navier–Stokes equations [9]. For an incompressible fluid, the continuity equation is given as: (24)$$\nabla \cdot u=\frac{\partial u_{x}}{\partial x}+\frac{\partial u_{y}}{\partial y}+\frac{\partial u_{z}}{\partial z}=0$$
where ∇ is the divergence and u is the fluid velocity. The Navier–Stokes momentum conservation equation of the motion for an incompressible fluid is expressed as:(25)$$\rho \frac{\partial u}{\partial t}+\rho \left(u\cdot \nabla \right)u=\nabla \cdot \left[-pI+\mu \left(\nabla u+{\left(\nabla u\right)}^{T}\right)\right]$$
where, ρ is the density of the fluid, μ is the viscosity of the fluid, p is the pressure, and I is the unit diagonal matrix. Finally, the piezo-membrane was fixed around the sides and a voltage waveform (shown in Figure 5) was applied to the upper electrode by considering the lower electrode as ground. The study of time dependence was used to simulate the model by considering the time ranging from 0 to 60 μs with a time step of 0.1 μs. The predicted pressure at the pressure chamber and velocity at the nozzle inlet were obtained and compared with the LEM results, as shown in Figure 6.

## 3. Results and Discussion

### 3.1. System Response

Ink with properties shown in Table 4 was used in the model. A voltage waveform with a 35 V amplitude, shown in Figure 5, with a 2 μs rising time ($t_{r}$), 6 μs duration time ($t_{d}$), and a 2 μs falling time ($t_{f}$) was applied to the piezo-membrane to investigate the pressure and velocity profiles at the pressure chamber and the nozzle inlet, respectively. Because of the complex geometry, the flow resistances of pressure chamber, restrictor, and nozzle presented in Table 3 were obtained from FEM simulations. The piezo-membrane deformed towards the pressure chamber during the rising edge of the voltage waveform, which generated a positive pressure of approximately 250 kPa (see Figure 6a), thereby pushing the ink toward the nozzle and both the inlet and the outlet restrictors. Due to this pressure, the meniscus with a convex surface was found at the nozzle with a pressure of 9.202 kPa, as shown in Figure 7. The membrane deformation stopped during the duration time, but there was still an exit pressure in the pressure chamber that was smaller in amplitude than the maximum pressure in the rising time, as shown in Figure 6a. During this time, the fluid slowly started to come toward the center of the pressure chamber from the nozzle exit and the restrictors with a gradually decreasing pressure amplitude. At the falling edge, the membrane deformed in the upper direction, generating a negative pressure in the pressure chamber (see Figure 6a), which sucked the fluid to the center of the chamber from all three directions. Due to this pressure, the meniscus with a concave surface was found at the nozzle with a pressure of −7.645 kPa, as shown in Figure 7.

After the ejection of first droplet at the nozzle exit, there is still a pressure perturbation at the pressure chamber called residual oscillations. These residual oscillations degrade the second drop size compared to the first droplet [20]. These oscillations depend on the viscosity of the ink and the geometry of the shape. In the non-recirculating inkjet printheads, one side is opened and the other is closed [7]; therefore, after the ejection of the first drop, the flow at the pressure chamber will experience a higher resistance, which can increase the residual oscillations. Recirculating inkjet printheads and the use of double restrictors can dampen these oscillations due to the reduced flow resistances at both the inlet and the outlet restrictors.

The pressure wave in the pressure chamber went to the nozzle inlet, which increased the nozzle velocity to eject the droplet at the nozzle exit. This velocity profile of approximately 4 m/s at the nozzle inlet is shown in Figure 6b. The velocity increased at the nozzle exit as the area decreased due to its conical shape. To verify the reliability of the presented lumped element model, a comparison of the pressure and velocity profiles between FEM simulations and LEM results was done; as shown in Figure 6, they almost match.

### 3.2. Analysis of the Helmholtz Resonance Frequency

When fluid is moving in the opposite direction in a two-hole resonator, it makes a resonance with a specific frequency called the Helmholtz resonance frequency [21], which is defined as:(26)$$f_{H}=\frac{1}{2\pi }\sqrt{\frac{c^{2}}{V_{p}}\left(\frac{A_{r}}{l_{r}}+\frac{A_{n}}{l_{n}}\right)}=\frac{1}{2\pi }\sqrt{\frac{1}{C_{p}}\left(\frac{1}{L_{r}}+\frac{1}{L_{n}}\right)}$$
where, c is the ink’s speed of sound; $V_{p}$ is the volume of the pressure chamber; $C_{p}$ is the acoustic compliance of pressure chamber; $L_{r}$ and $L_{n}$ are the acoustic inductance of the restrictor and the nozzle, respectively; and $A_{r}$, $A_{n}$, and $l_{r}$, $l_{n}$ are the area and length of the restrictor and the nozzle, respectively. The subscripts *p*, *r*, and *n* denote the pressure chamber, restrictor, and nozzle, respectively. Generally, a recirculating inkjet printhead contains three holes: the inlet, outlet, and nozzle. Depending on the polarity, when the rising or the falling edge of the voltage waveform is applied to the membrane, it pushes the ink from the pressure chamber to three directions or sucks the ink from three directions to the center of the pressure chamber in anti-phase to each other. This process repeats at the Helmholtz frequency. In our proposed D-RPIP design, there are two restrictors at the inlet and two at the outlet. Therefore, the acoustic inductance in these restrictors has to be added to obtain correct estimated calculations of the Helmholtz resonance frequency. For the D-RPIP design, Equation (26) could be written as: (27)$$f_{H}=\frac{1}{2\pi }\sqrt{\frac{1}{C_{p}}\left(\frac{1}{L_{ri1}}+\frac{1}{L_{ri2}}+\frac{1}{L_{ro1}}+\frac{1}{L_{ro2}}+\frac{1}{L_{n}}\right)}$$
By putting the values of all these acoustic inductances and compliances, the Helmholtz frequency was found to be 598 kHz. The Helmholtz frequency was also found using the lumped element model, as shown in Figure 8, giving a value of approximately 500 kHz, which is close to calculated frequency. The Helmholtz resonance frequency was analyzed by changing the width and depth of the pressure chamber and the restrictor. From Equation (27), it is seen that the Helmholtz resonance frequency depends on the acoustic compliance of the pressure chamber and the acoustic inductance of the restrictors and the nozzle. The acoustic compliance of the pressure chamber can be given by the following relation [7]:(28)$$C_{p}=\frac{A_{p}\times l_{p}}{\rho \times \left(C_{eff}{}^{2}\right)},C_{eff}=\frac{c}{\sqrt{1+\beta }},\beta =\left(\frac{\Delta V_{0}}{P}\right)\left(\frac{\rho c^{2}}{V_{0}}\right)$$
where $A_{p}$, $l_{p}$, and $V_{0}$ are the area, length, and volume of the pressure chamber, respectively; ρ is the density of the fluid; c is the actual speed of sound; and $C_{eff}$ is the effective speed of sound. Figure 9 shows the effect of changing the design parameters on the Helmholtz resonance frequency. By increasing the width of the pressure chamber ($w_{p}$), the width of the membrane also must be increased. As a result, the ratio of initial chamber volume ($V_{0}$) and displaced volume ($\Delta V_{0}$) increased. Due to the increase in $\frac{\Delta V_{0}}{V_{0}}$, the acoustic compliance ($C_{p}$) in the pressure chamber increases, which led to the decrease of the Helmholtz resonance frequency, as shown in Figure 9a. In the case of increasing the restrictor width ($w_{r}$), the Helmholtz resonance frequency increased, as shown in Figure 9b, because the acoustic inductance decreased by increasing the width of the restrictor. A similar held for the pressure chamber depth ($d_{p}$), and the restrictor depth ($d_{r}$), variation. Namely, the Helmholtz resonance frequency decreased in the case of increasing the pressure chamber depth (see Figure 9c), and increased in the case of increasing the restrictor depth (see Figure 9). In the case of the increasing length of both the pressure chamber ($l_{p}$) and restrictor ($l_{r}$), the Helmholtz resonance frequency decreased, as shown in Figure 9e,f. A comparative analysis between the pressure chamber and restrictor was needed in the case of depth variation, since they usually have the same depth. From our analysis, it was concluded that the increment rate of the Helmholtz resonance frequency due to the increase in the depth of the restrictor was higher than the decrement rate of the same frequency due to the increase in the depth of the pressure chamber, as shown in Figure 10.

### 3.3. Analysis of Pressure and Velocity

In our study, the effect of changing the width and depth of the pressure chamber and the restrictor on the pressure at the pressure chamber and velocity at the nozzle inlet was investigated. The conclusions are as follows. With the increase in the restrictor width and depth, both the pressure and the velocity decreased, as is shown in Figure 11. By increasing the width of the pressure chamber, the pressure and the velocity increased, as seen in Figure 12a,b, respectively. By contrast, by increasing its depth, the pressure (see Figure 12c) and velocity (see Figure 12d) decreased slightly. By increasing the length of the pressure chamber, both the pressure at the pressure chamber and velocity at the nozzle inlet were slightly increased (see Figure 13a,b, respectively). On the other hand, by increasing the length of the restrictor, both the pressure at the pressure chamber and velocity at the nozzle inlet decreased, as shown in Figure 13c,d, respectively.

As shown in Figure 11a,c, Figure 12a, and Figure 13a,c, there were residual oscillations after the ejection of the first droplet. These oscillations degraded the volume of the second droplet. On the basis of these unwanted oscillations, we finalized the optimized device dimensions presented in Table 1. With the optimized device dimensions, the residual oscillations were smaller than the other device dimensions. In addition, it took less time to dampen these oscillations in the case of optimized parameters, as shown in the Figure 11, Figure 12 and Figure 13.

From our analysis, it was also concluded that the restrictor depth variation had a greater effect than the pressure chamber depth variation. As shown in Figure 14a, until the depth of 70 μm, the effect of increasing the depth of the restrictor on the pressure at the pressure chamber (keeping the pressure chamber depth constant) showed a larger decrement rate than the effect of increasing the depth of the pressure chamber on the pressure at the pressure chamber (keeping the restrictor depth constant). After the depth of 70 μm, it showed a similar decrement rate. In addition, in the case of the velocity, the restrictor depth variation effect was higher than the pressure chamber depth variation effect (see Figure 14b). This shows that the restrictor was more sensitive than the pressure chamber; therefore, more attention is needed to fabricate the restrictor using micro-electro-mechanical systems (MEMS) technology without fabrication error because a small error can deviate the device performance from the required results.

### 3.4. Waveform Effect on the System’s Response

The pressure response in the pressure chamber was analyzed by changing the voltage waveform’s rising, falling, and duration times. By increasing the rising and falling times simultaneously, the pressure decreased, as shown in Figure 15. In addition, the negative pressure from the falling edge of the voltage waveform was delayed. In our analysis, we used a push–pull mode actuation method, so there was no change in the maximum pressure the duration time of voltage waveform was increased with a fixed rising and falling time, as shown in Figure 16. Increasing the duration time affected the negative pressure in the form of pressure amplitudes and time delays. With the duration times of 4 μs, 6 μs, and 8 μs, the amplitudes of the minimum pressure generated by the falling edge were −285, −275, and −267 kPa, respectively, as shown in Figure 16. This decrease in the amplitude of the minimum pressure was due to the fact that when the duration time increased, the membrane vibration decreased, thereby decreasing the amplitude of the negative pressure generated by the falling edge.

### 3.5. Hydro-Acoustic Cross-Talk Effect

In the inkjet printhead systems, cross-talk happens whenever the activation of the nozzle or the number of nozzles disturbs the neighboring nozzles, which can affect the drop velocity and volume. There are two main cross-talks, namely structural and hydro-acoustic, that occur during the printhead operation. Structural cross-talk happens due to the mechanical connection between the nozzles, while hydro-acoustic cross-talk occurs when all the nozzles are connected to a common flow channel through the restrictors. When a single nozzle is activated, it generates back pressure perturbations in the common flow path at the restrictor inlets of the neighboring nozzles. Hydraulic cross-talk can be reduced by reducing the acoustic impedance of the common flow path. To reduce the acoustic impedance, two methodologies have been presented in the literature: one is to increase the height of the common flow path, while the other one is to increase the acoustic compliance by covering the common ink supply with foil [22].

In our study, we analyzed the hydro-acoustic cross-talk of our proposed D-RPIP design using lumped modeling. The connection of multiple nozzles in an array with the common flow channel is shown in Figure 17. A total of 80 nozzles was used in the array. Using an electric-fluid analogy, the fluid contained in the common flow channel was modeled as an acoustic capacitance, inductance, and resistance. The cross-talk effect was estimated in the common flow path at the restrictor inlet of the right-hand side printheads starting from the central printhead. The pressure obtained from the lumped modeling of the individual nozzle was used as an input to the LEM for the cross-talk. Using Equations (7) and (8), the first-order differential equations of the state variables from Figure 17 have were obtained as follows:(29)$$\frac{dx_{1}}{dt}=\frac{1}{C_{a}}\left(\frac{P_{in}}{R_{in}}-2\vec{x_{2}}\right)$$
(30)$$\frac{d\vec{x_{2}}}{dt}=\frac{1}{L_{a}}\left(x_{1}-R_{a}\vec{x_{2}}-x_{3}\right)$$
(31)$$\frac{dx_{3}}{dt}=\frac{1}{C_{a}}\left(\vec{x_{2}}-\frac{x_{3}}{R_{in}}-\vec{x_{4}}\right)$$
(32)$$\frac{d\vec{x_{4}}}{dt}=\frac{1}{L_{a}}\left(x_{3}-R_{a}\vec{x_{4}}-x_{5}\right)$$
where $L_{a}=\frac{\rho l}{A}$, $C_{a}=\frac{V_{0}}{\rho c^{2}}$, and $R_{a}=\frac{8\mu \pi l}{A^{2}}$ are the acoustic inductance, the capacitance, and the resistance in the common flow channel, respectively. As before, ρ, c, and μ are the ink’s density, speed of sound, and viscosity, respectively. Next, $V_{0}$, *l*, and *A* are, respectively, the volume, length, and cross-sectional area of the channel in which ink is flowing. Finally, $P_{in}$ is the input pressure to this model, which was obtained from the LEM of individual printhead. The total acoustic resistance of the double-inlet restrictors that made this a parallel configuration is represented by $R_{in}$, defined as:(33)$$R_{in}=\frac{R_{ri1}\times R_{ri2}}{R_{ri1}+R_{ri2}}$$
From Equations (29) to (32), we obtained: $A=\left[\begin{array}{cccccccc} 0 & -2/C_{a} & 0 & 0 & 0 & . & . & .\\ 1/L_{a} & -R_{a}/C_{a} & -1/L_{a} & -1/C_{a} & 0 & . & . & .\\ 0 & 1/C_{a} & -1/C_{a}R_{in} & -R_{a}/L_{a} & 0 & . & . & .\\ 0 & 0 & 1/L_{a} & -1/L_{a} & 0 & . & . & .\\ . & . & . & . & . & . & . & .\\ . & . & . & . & . & . & . & .\\ . & . & . & . & . & . & . & . \end{array}\right]$
and $B=\left[\begin{array}{c} 1/C_{a}R_{in}\\ 0\\ 0\\ .\\ .\\ .\\ 0 \end{array}\right]$.

For the last nozzle, the equation is given below: (34)$$\frac{dx_{2N-1}}{dt}=\frac{1}{C_{a}}\left(x_{2N-2}-\frac{x_{2N-1}}{R_{in}}-\frac{x_{2N-1}}{R_{a}}\right)$$
The effect of the number of simultaneously activated neighboring nozzles on the single central nozzle is shown in Figure 18a. When only a single central nozzle was activated, a maximum back pressure of 9.65 kPa was found in the common flow channel at the restrictor inlet of the activated single central nozzle. When the number of activated nozzles were increased, the pressure of the central nozzle at the common flow channel also increased, which produced a saturation level after the activation of almost 10 nozzles, as shown in Figure 18b. This means that the cross-talk effect of neighboring nozzles on the central activated nozzle became constant when the number of activated neighboring nozzles increased. The cross-talk effect of a single activated nozzle on the neighboring non-activated nozzles was also investigated. It was shown that the pressure perturbation gradually decreased when the distance between the activated nozzle and neighboring non-activated nozzles was increased, as shown in Figure 19.

## 4. Conclusions

In this research, we proposed a recirculating MEMS-based piezo-driven inkjet printhead for digital textile printing applications. Two-port lumped element modeling was done by defining eleven state variables to solve the first-order differential equations of the state space model. The analysis led to the following conclusions.

The lumped element model for a recirculating inkjet printhead having double restrictors at the inlet and outlet was presented. Using the LEM, four major output parameters—jetting pressure at the pressure chamber, the velocity at the nozzle inlet, meniscus pressure and Helmholtz resonance frequency—were obtained. The jetting pressure at the pressure chamber and the velocity at the nozzle inlet were compared with the FEM simulations, which showed perfect matching. This shows the accuracy of our proposed lumped modeling for recirculating inkjet printheads. The effect of the variation of major design parameters on the jetting pressure, the velocity, and the Helmholtz resonance frequency were analyzed, and the device dimensions, which showed reduced residual oscillations, were finalized and presented in Table 1. As the depth of the pressure chamber and restrictor was same, the effect of their variation on the resulted parameters were investigated and compared. It was concluded that the effect of the depth variation of the restrictor on the Helmholtz resonance frequency, jetting pressure, and velocity was greater than the effect of the pressure chamber depth variation.

After obtaining the jetting pressure at the pressure chamber from the optimized device dimensions, the jetting pressure was further investigated by changing the rising, falling, and duration time of the voltage waveform. The changing of rising and falling times simultaneously had a larger effect on the pressure at the pressure chamber compared to the effect of duration time.

Moreover, the hydro-acoustic cross-talk of multi-nozzle printheads was also investigated using the LEM. The cross-talk effect of multi-activated nozzles on the single central activated nozzle increased when the number of neighboring activated nozzles was increased. The cross-talk effect of the central activated nozzle on the neighboring non-activated nozzles was also investigated, and it was found that the cross-talk effect gradually decreased when the distance between the central activated nozzle and the neighboring non-activated nozzles increased.

## Figures and Tables

**Figure 1 micromachines-10-00757-f001:**
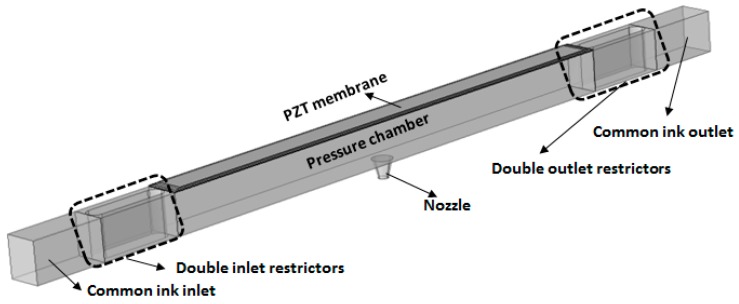
3D view of double-restrictor recirculating piezoelectric inkjet printhead (D-RPIP) design.

**Figure 2 micromachines-10-00757-f002:**
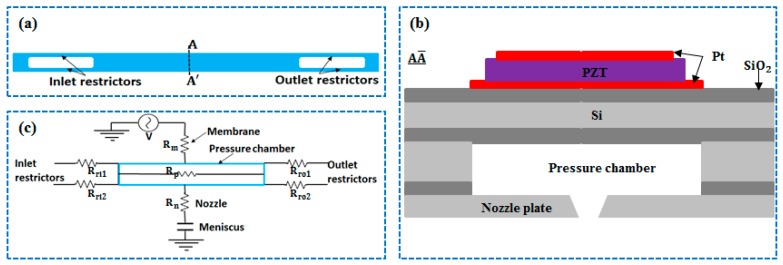
(**a**) Top view, (**b**) cross-sectional view, and (**c**) electrical representation of a D-RPIP design.

**Figure 3 micromachines-10-00757-f003:**
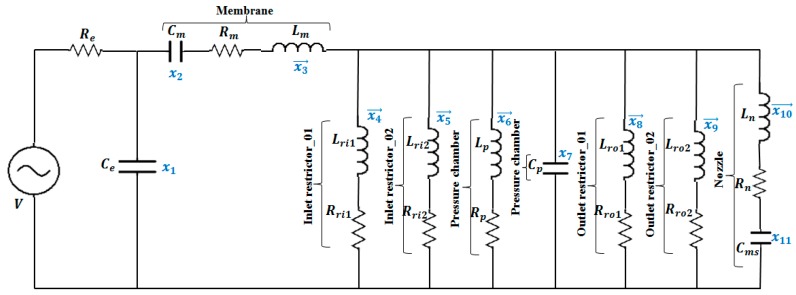
Equivalent circuit model of a D-RPIP design.

**Figure 4 micromachines-10-00757-f004:**
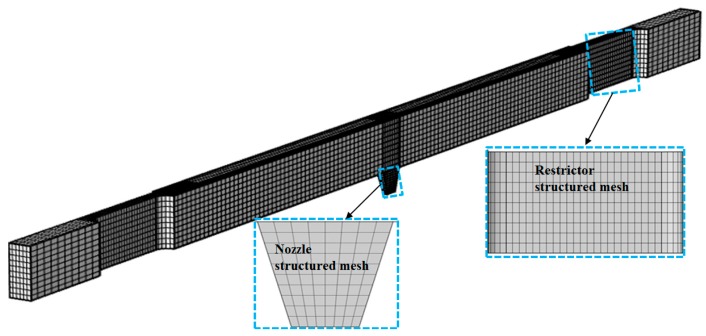
Meshed 3D model of half of the proposed recirculating inkjet printhead.

**Figure 5 micromachines-10-00757-f005:**
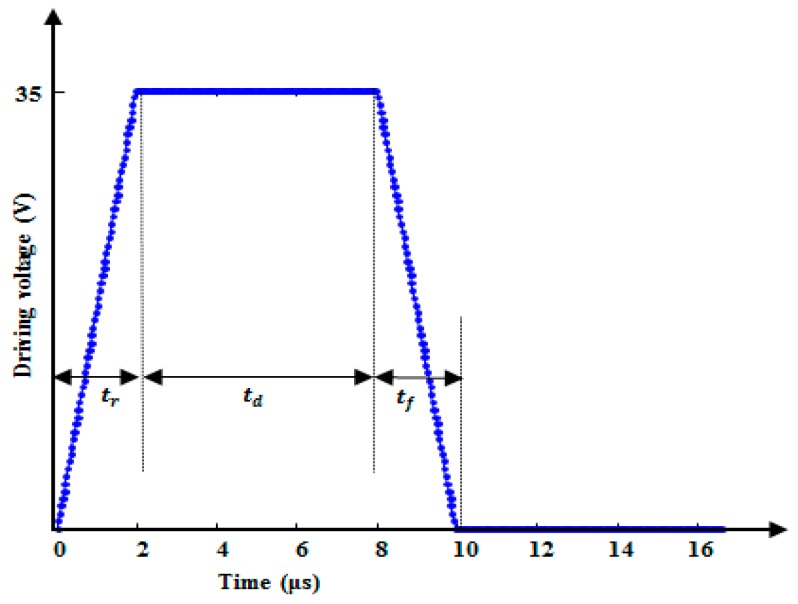
Input voltage waveform applied in both the LEM and FEM simulations.

**Figure 6 micromachines-10-00757-f006:**
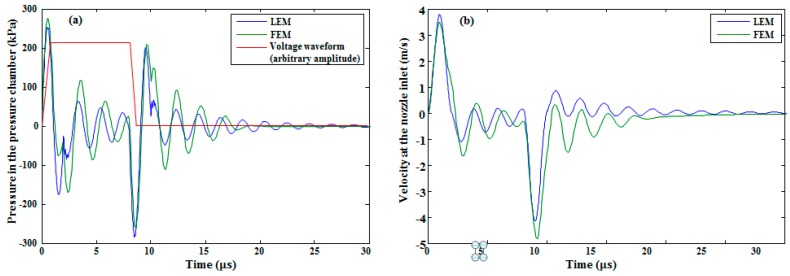
(**a**) Pressure at the pressure chamber and (**b**) velocity at the nozzle inlet.

**Figure 7 micromachines-10-00757-f007:**
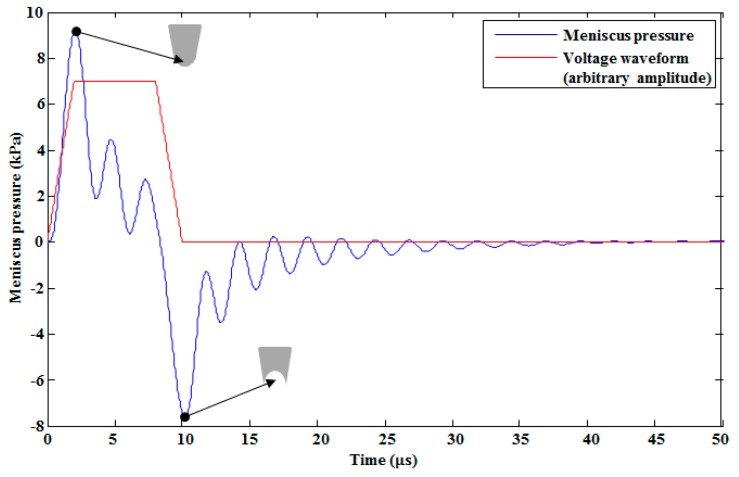
Meniscus pressure on the rising and falling edges of the voltage waveform.

**Figure 8 micromachines-10-00757-f008:**
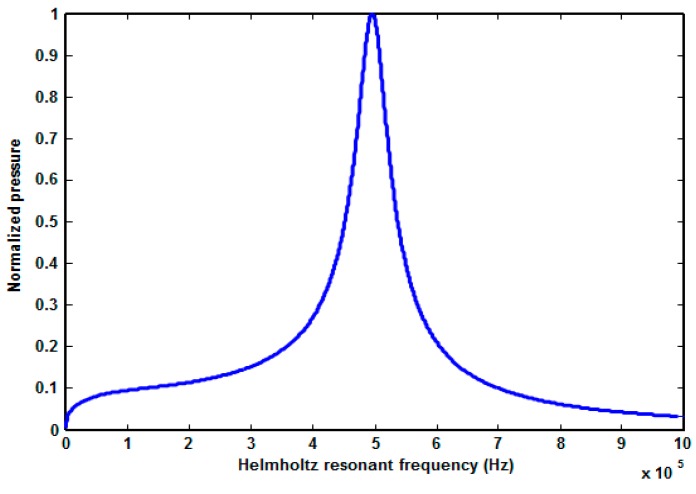
Helmholtz resonant frequency using the LEM.

**Figure 9 micromachines-10-00757-f009:**
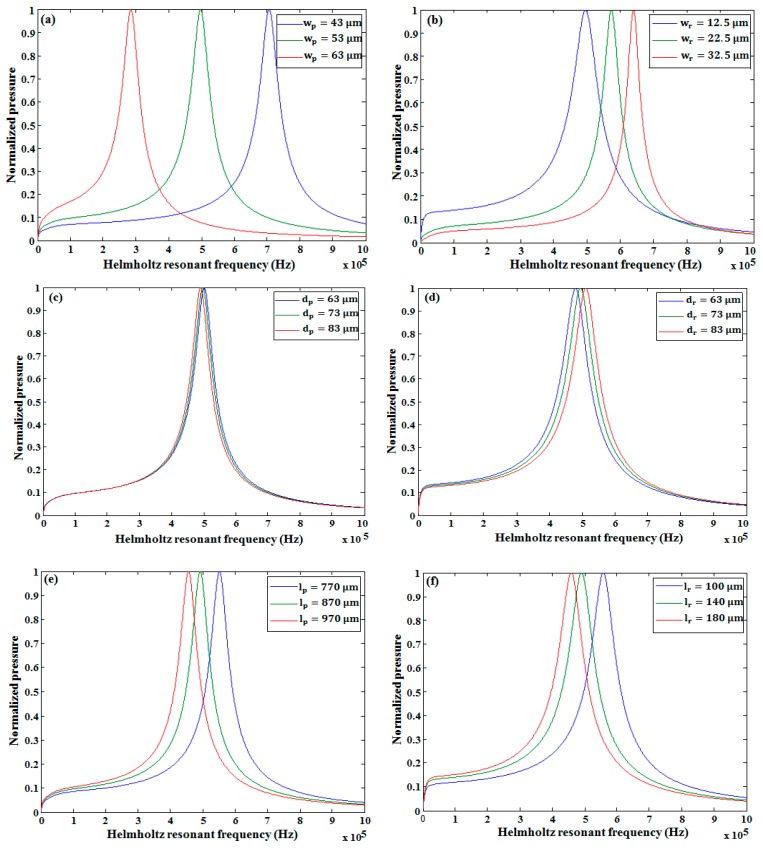
Change of the Helmholtz resonance frequency by varying the (**a**) pressure chamber width, (**b**) restrictor width, (**c**) pressure chamber depth, (**d**) restrictor depth, (**e**) pressure chamber length, and (**f**) restrictor length.

**Figure 10 micromachines-10-00757-f010:**
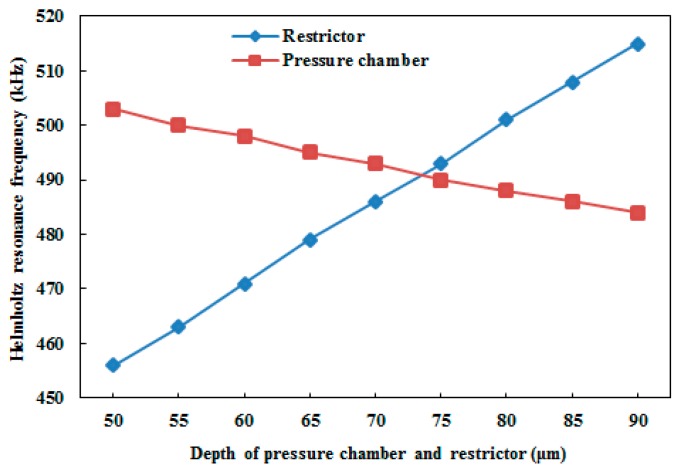
Comparative analysis of the effect of the depth variation of the pressure chamber and the restrictor on the Helmholtz resonance frequency.

**Figure 11 micromachines-10-00757-f011:**
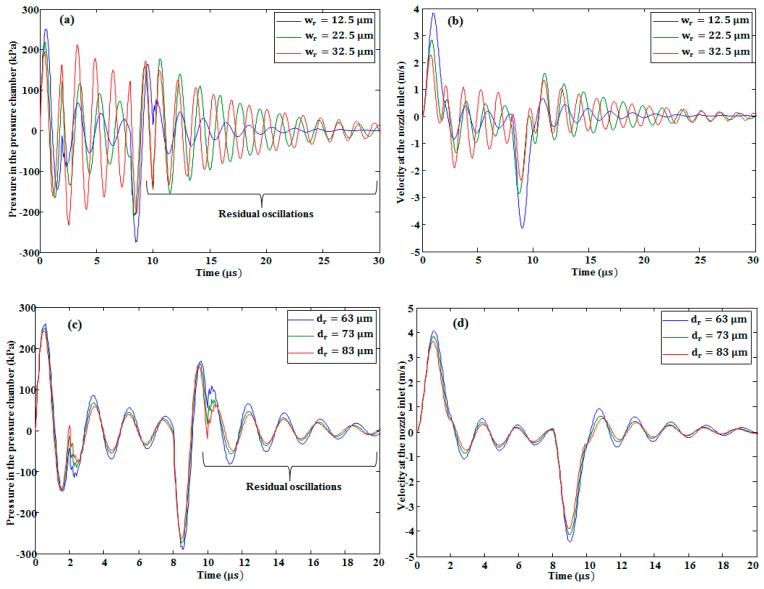
Restrictor width variation effect on (**a**) pressure and (**b**) velocity. Restrictor depth variation effect on (**c**) pressure and (**d**) velocity.

**Figure 12 micromachines-10-00757-f012:**
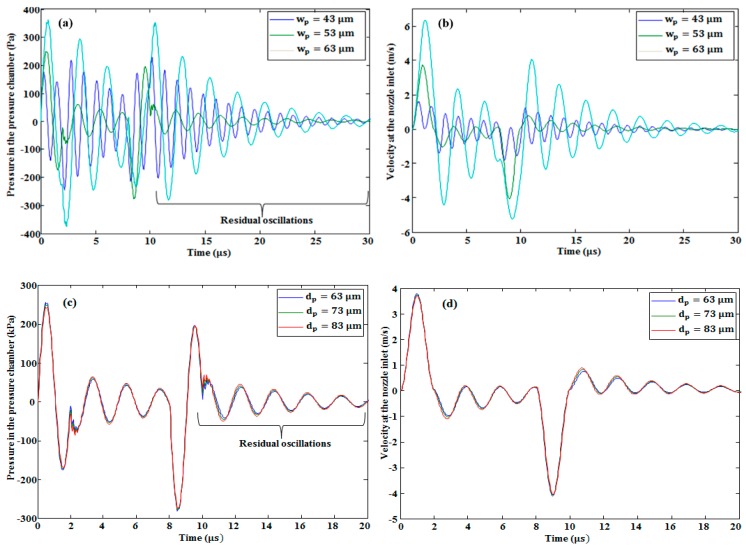
Effect of pressure chamber width variation on (**a**) pressure in the pressure chamber and (**b**) velocity at the nozzle inlet. Effect of pressure chamber depth variation on (**c**) pressure and (**d**) velocity.

**Figure 13 micromachines-10-00757-f013:**
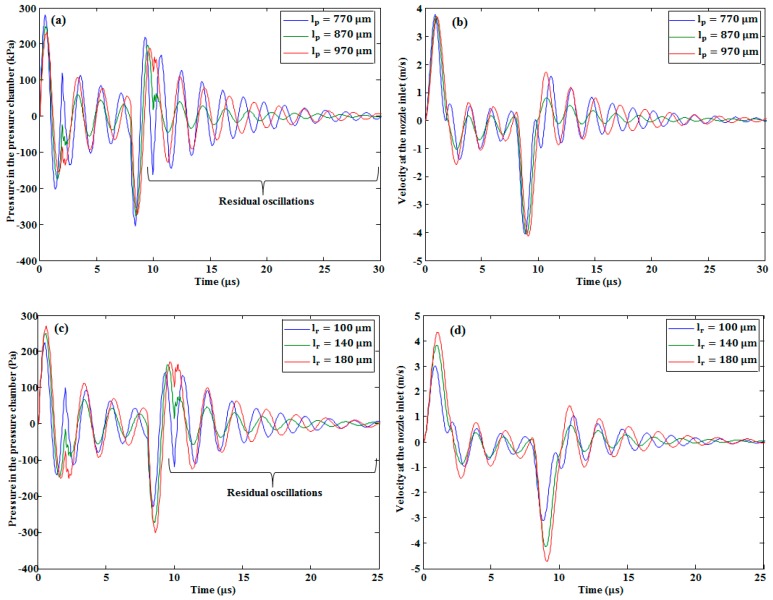
Effect of pressure chamber length variation on (**a**) pressure at the pressure chamber and (**b**) velocity at the nozzle inlet. Effect of restrictor length variation on (**c**) pressure and (**d**) velocity.

**Figure 14 micromachines-10-00757-f014:**
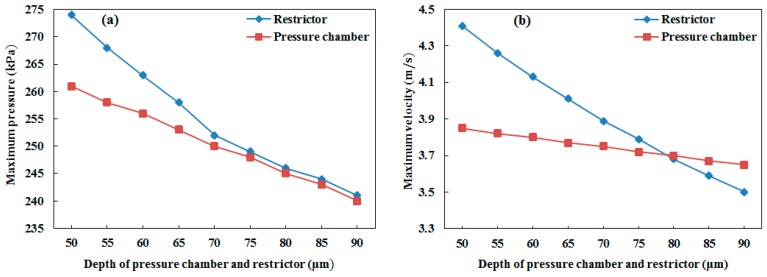
Restrictor and pressure chamber depth variation effect on (**a**) maximum pressure at the pressure chamber and (**b**) maximum velocity at the nozzle inlet.

**Figure 15 micromachines-10-00757-f015:**
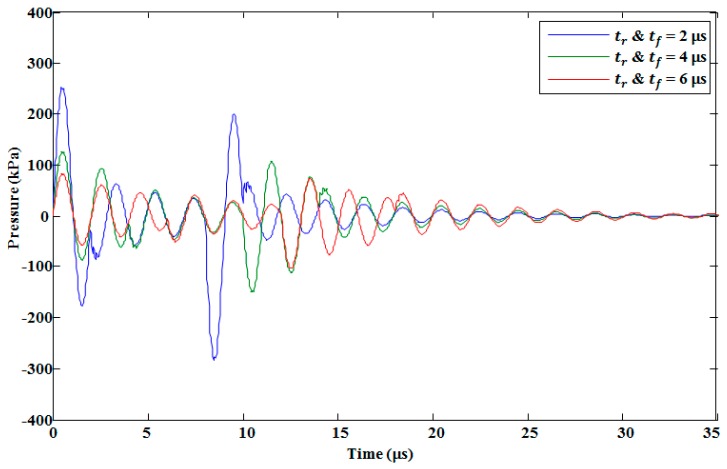
Effect of simultaneously changing the rising and falling time of the trapezoidal voltage waveform on the pressure in the pressure chamber.

**Figure 16 micromachines-10-00757-f016:**
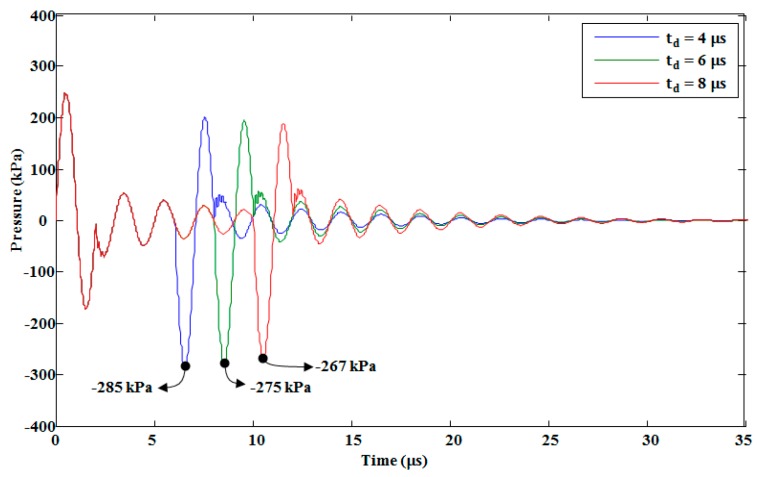
Effect of changing the duration time of the trapezoidal voltage waveform on the pressure in the pressure chamber.

**Figure 17 micromachines-10-00757-f017:**
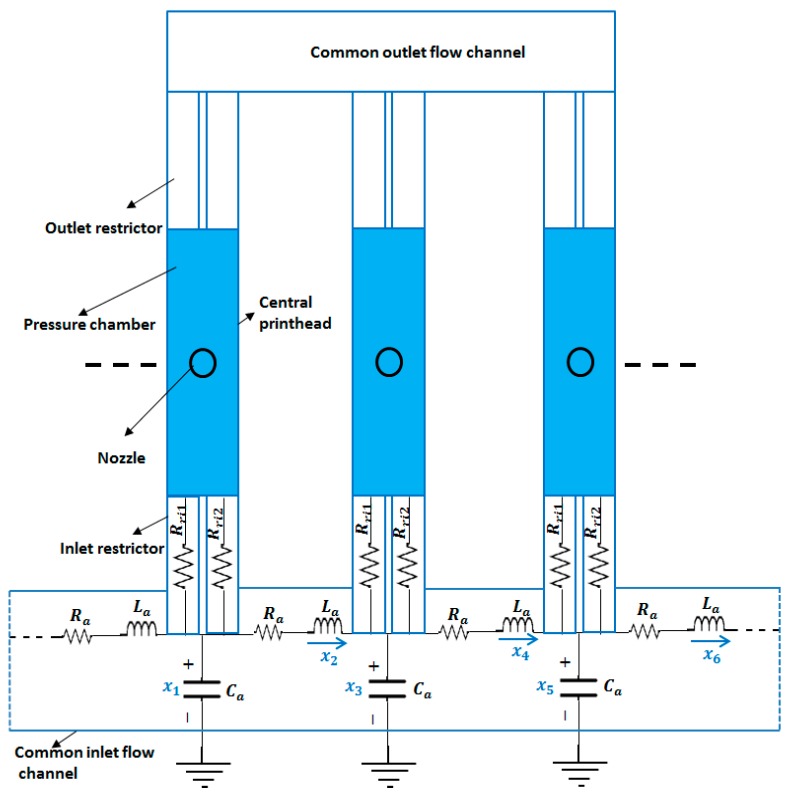
Array of multiple printheads with an equivalent electrical circuit model at the common inlet flow channel.

**Figure 18 micromachines-10-00757-f018:**
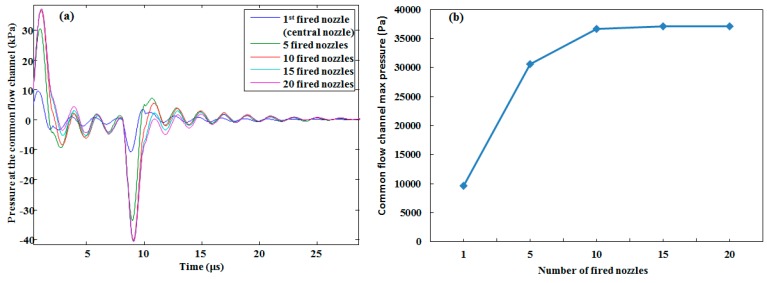
(**a**) Pressure perturbation at the common flow channel of the central activated nozzle when only the central nozzle was fired and with the given numbers of simultaneously fired nozzles. (**b**) Saturation of the pressure perturbation at the common flow channel when the number of fired nozzles was increased.

**Figure 19 micromachines-10-00757-f019:**
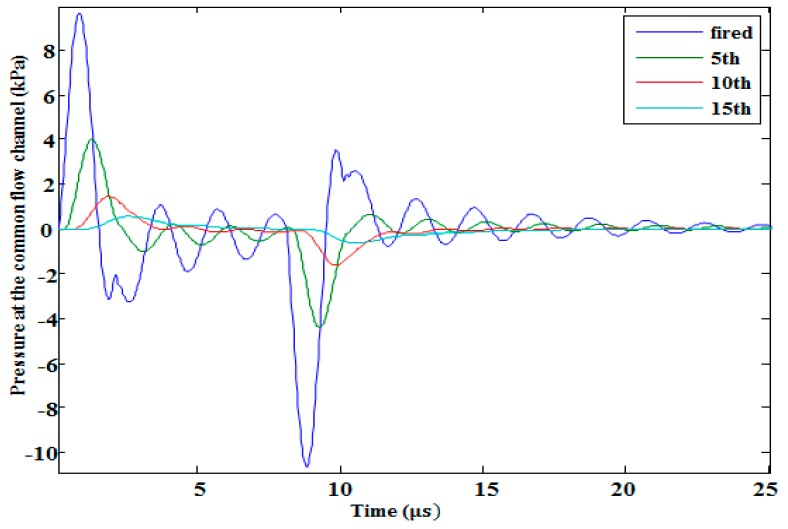
Cross-talk effect of the central activated nozzle on the neighboring non-activated nozzles.

**Table 1 micromachines-10-00757-t001:** Device dimensions.

Pressure Chamber			Restrictor			Nozzle		
Length (μm)	Width (μm)	Depth (μm)	Length (μm)	Width (μm)	Depth (μm)	Length (μm)	Inlet Radius (μm)	Outlet Radius (μm)
870	53	73	140	12.5	73	33	16	8

**Table 2 micromachines-10-00757-t002:** Piezo-membrane dimensions.

PZT		Platinum			Silicon		SiO_2_	
Width (μm)	Thickness (μm)	Width (μm)	Thickness (μm)		Width (μm)	Thickness (μm)	Width (μm)	Thickness (μm)
			Upper	Lower				
44	2	44	0.11	0.15	53	1.1	53	0.5

**Table 3 micromachines-10-00757-t003:** Parameters used in lumped element modeling. FEM—finite element modeling.

Parameters		Formulas and Simulations	Values
Pressure chamber	$C_{p}\left(m^{4}\cdot s^{2}\cdot {\mathrm{kg}}^{-1}\right)$	$\frac{A_{p}\times l_{p}}{\rho \times C_{eff}{}^{2}}$	$3.082\times 10^{-21}$
	$L_{p}\left(\mathrm{kg}\cdot m^{-4}\right)$	$\frac{\rho \times l_{p}}{A_{p}}$	$2.478\times 10^{8}$
	$R_{p}\left(\Omega \right)$	FEM	$1.027\times 10^{13}$
Nozzle	$L_{n}\left(\mathrm{kg}\cdot m^{-4}\right)$	$\frac{\rho \times l_{n}}{A_{n}}$	$4.521\times 10^{7}$
	$R_{n}\left(\Omega \right)$	FEM	$2.05\times 10^{13}$
Restrictor	$L_{r}\left(\mathrm{kg}\cdot m^{-4}\right)$	$\frac{\rho }{A_{r}}\left(l_{r}+0.48\sqrt{A_{r}}\right)$	$1.865\times 10^{-4}$
	$R_{r}\left(\Omega \right)$	FEM	$3.09\times 10^{13}$

**Table 4 micromachines-10-00757-t004:** Ink properties used in lumped element model (LEM) and FEM simulations.

Density (kg·m^−3^)	Surface Tension (N/m)	Viscosity (Pa·s)
1102	34.68 × 10^−3^	5.95 × 10^−3^

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
