# Peer review of "Design and Characteristic Analysis of a MEMS Piezo-Driven Recirculating Inkjet Printhead Using Lumped Element Modeling"

_micromachines, 2019, doi:10.3390/mi10110757_

Round 1
Reviewer 1 Report
Dear Authors,
I did read your manuscript with great interest. Please consider the attached comments.
Your paper is an exercise in Lumped Element Method applied to a hypothetical print head. It has been done in the right way as it complies with the outcomes of a standard software package. I really miss the confrontation with reality, with the operation of a real print head.

Author Response
Authors responses to 1st reviewer comments
Comments to “Design and Characteristic Analysis of MEMS Piezo-Driven Recirculating Printhead using Lumped Element Modelling”by M.A. Shah, D-G Lee and S. Hur.
Micromachines 2019 for peer review.
The authors thank the reviewer and the editor for providing constructive comments and suggestions to improve the scientific quality and readability of the manuscript. The author’s responses to the reviewer’s comments are below:
Comments:
Abstract: it is an exercise in modelling rather than really building a print head?
Author’s response: We thanked the reviewer for asking the clarification. It is the modelling for a real print head to be fabricated and developed.
25: Do you mean hundreds to thousands of nozzles or indeed hundreds of thousands of nozzles?
Author’s response: We are thankful for the reviewer’s question. We have changed it to hundreds or thousands of nozzles.
41: next to the book of Hoath et al, you can also refer to
Inkjet Technology edited by I.M. Hutchings and G.D. Martin, Wiley 2013 Inkjet-based Micromachining edited by J.G. Korvink, P.J. Smith and D-Y Shin Wiley-VCH 2012 Handbook of Industial Inkjet Printing, two volumes edited by W. Zapka, Wiley-VCH 2018
Author’s response: We thanked the reviewer for referring these informative books. The above mentioned books are cited next to the book of Hoath et al.
47: Lumped element modelling means a finite element method where all the mass of the element is concentrated in one point. This point is connected to other elements by springs and dampers?
Author’s response: Lumped element modelling and finite element method are two different approaches we performed for our inkjet printhed. Finite element method is the numerical simulations technique which we performed in COMSOL Multiphysics software. We have added this FEM approach in our paper also. We did this simulation just to get the pressure and velocity profiles at the pressure chamber and at the nozzle inlet and compared it with the proposed lumped element modeling results to validate our proposed lumped element modeling. We further investigate our inkjet head using lumped modeling because it is an easy and time consuming approach than the finite element modeling to analyze all the dimensions of a design. In this Lumped element modeling approach we distribute the physical system into a topology consisting of discrete entities. The entities are represented in the form of an electrical circuit and the fluid resistance, compliance, and fluid mass are considered as electrical resistance, capacitance and inductance respectively.
48: What do you mean with two port LEM? When I understand it correctly extensive finite element modelling is used to derive the properties of an one dimensional description of e.g. a piezo?
Author’s response: Two port network is a pair of two terminal electrical network in which, current enters through one terminal and leaves through another terminal of each port. Two port network representation is shown in the below figure, in which 1 is first port and 2 is the second port. We have already explained our approach for LEM in the above answer.
56: One end closed at the nozzle? The nozzle is neither an open end or a closed end (see J.F. Dijksman Design of Piezo Inkjet Print Heads, Wiley-VCH, 2018 chapters 5.2.1 and 5.2.2)
Author’s response: We thanked the reviewer for valuable suggestion. We have now rephrased the sentence.
67: Please refer to the re-circulating heads of Xaar and Konica-Minolta of already existing print heads with recirculation in the market! See e.g.: “Shear-mode Piezo Inkjet Head with Two Recirculating Paths”, Hikaru Hamano, Taishi Shimizu, Takuma Shibata, Yasuhiko Suetomi, Kazuki Hiejima, and Yusuke Kuramochi, (all from Konica Minolta), Proceedings Printing for Fabrication 2019, pp 173-176.
Author’s response: We thanked the reviewer for referring this paper. We read the paper and cited in our paper.
80: Rephrase sentence
Author’s response: We thanked the reviewer for valuable suggestion. We rephrased the sentence in our updated file of the paper according to the above suggested paper in the above comment of the reviewer.
111: Figure 2c is not clear, all springs should be connected by nodal points, and there is no nodal point between the pressure chamber and nozzle???
Author’s response: Figure 2c is a general representation of fluid-electric analogy circuit. The nodal analysis using KCL is done on some nodes of figure 3 of the paper, for example see below figure for equation 9 in the paper.
124: Here the assumption should be mentioned that is used here: namely that the fluid volume displaced is just half sphere and that with such a volume displacement the capillary pressure just is defined by a spherical dome of height equal to the nozzle radius. For small fluid displacements the meniscus generates a kind of linear stiffness for larger displacements the capillary pressure does not change while the volume displaced increases accordingly (see J.F. Dijksman Design of Piezo Inkjet Print Heads, Wiley-VCH, 2018 chapter 3.6).
Author’s response: According to the reviewer’s valuable suggestion, we mentioned it in our updated file.
136: There are two ends of the resistor should 0.48 read 0.96?
Author’s response: In our design, the only one end is a flanged end at the reservoir side, so is the reason we took 0.48.
147: You refer to FEM calculations to determine the flow resistances in the system. Are these calculations on static flow resistance: the steady flow of a viscous fluid through a complicated shaped channel or did you take into account the frequency dependence as well?
Author’s response: It is the steady flow of a viscous fluid through a complicated shaped channel. We have used the below formula:
R=delta(P)/volume flow rate(Q), where Q was found from COMSOL FEM simulations of these micro-channels.
144: The inertia length of the nozzle and viscous drag depend on its conical shape (see J.F. Dijksman Design of Piezo Inkjet Print Heads, Wiley-VCH, 2018, chapter 3.1.3)
Author’s response: For finding the fluid inertance, we have used the averaged value of the radius of a conical nozzle similar to the approach presented in the reference [Ref] below:
[Ref] Wang, J.; Huang, J.; Peng, J. Hydrodynamic response model of a piezoelectric inkjet print-head. Sensors Actuators, A Phys., 2019, 285, 50–58.
148: see previous remark
188: The solution is found by stepping through the time domain, starting from idling situation??
Author’s response: Yes it is starting from the idling situation.
193: The choice of the wave form is of ultimate importance. Why did you start with 2 µs rise time, 6 µs dwell time and 2µs shut-off time? What is the reasoning? Did you a scan through the “dwell time” domain to find the optimal setting? Or did you take into account limitations of the drive electronics? What about the next (second) droplet. Is the second droplet needed for extra volume? From your results the optimal pulse seems to be a square pulse with a up time of 1 µs.
Author’s response: This waveform of 2um rise time, 6um dwell time and 2um shut-off-time is the optimum waveform by considering the residual oscillation. We have already checked it from 1us of rising and falling time. Although, decreasing the rise and falling time to 1us increases the pressure and velocity, however there come residual oscillations after the ejection of the first drop. So is the reason we have taken the 2-6-2 rising-dwell-falling waveform.
199: 2500 kPa should read 250 kPa.
Author’s response: We thanked the reviewer for pointing the typing mistake. We have now changed it to 250 kPa.
201: The meniscus pressure is given by (see formula 1) by 2s/rn . For the data entered this value equals 8.67 kPa (valid for a concave as well as a convex meniscus). Why are the values you find from your calculations slightly different.
Author’s response: We thankful to the reviewer for asking this clarification. As in the LEM method, dynamic effects occurring while the formula used is just steady effect, that is the reason of a slight difference.
210: You refer to droplets. I guess that the wave form you have used will most probably not produce droplets. The pressure in the pump chamber is just ¼ of a bar and the velocity in the nozzle 3.5 m/s (see J.F. Dijksman Design of Piezo Inkjet Print Heads, Wiley-VCH, 2018, chapters 7.2.2 and 7.3.2)
Author’s response: We thankful to the reviewer for asking this clarification. This waveform is just to analyze our LEM model results. The velocity of 3.5m/s is at the nozzle inlet, therefore it will not be considered as a jetting velocity. The velocity at the nozzle exit will increase four times (14 m/s) of the inlet velocity due to the decrease in its area (nozzle inlet radius=16um, nozzle outlet radius=8um).
From the reviewer’s mentioned book theory, the jetting velocity should be greater than the capillary velocity, that is: , where , where is the nozzle radius at the nozzle exit. By putting the ink properties presented in table 4 of our paper, and putting nozzle exit radius of 8um, we can get, , which is lesser than the jetting velocity . So it is possible to jet the droplets at the nozzle exit using our analysis.
220: Is the COMSOL calculation done in the time domain, give some detail about time step and so.
Author’s response: Yes, the COMSOL simulation is done in the time domain. The study of time dependent is used to simulate the model by considering the time ranging from 0 to 60 with a time step of 0.1 .
256: In formula 26 you use P, what is P? Pressure? The ratio of DV0 and P must be obtained by FEM analysis?
Author’s response: The P is the applied pressure. Yes we have obtained the deformed volume (DV0) by FEM using COMSOL software.
366: 80 print heads or 80 nozzles
Author’s response: We thank the reviewer for pointing it out. We have changed it to 80 nozzles.
409: Interesting is that the recirculation is only mentioned by the presence of an inlet and an outlet. The nozzle placed symmetrically in between. There is no effect of the circulating flow of the jetting characteristics?
Author’s response: We thank the reviewer for asking this practical question. Indeed the nozzle, which is placed in the center of the design symmetrically, will have effect but as the recirculation phenomena is more concerned with the inlet and outlet restrictors, so we have considered on these two.
412: The four major output parameters most probably do not end up with droplet formation at all. What is the reason giving these numbers?
Author’s response: We removed the numbers from the abstract and conclusions in the updated file.

Reviewer 2 Report
The authors present a modeling approach for analyzing the recirculating piezo-driven MEMS based inkjet printhead. The lumped element model is developed to investigate the effect of various design parameters on the responses of the system. The cross-talk effect is also studied with the proposed model. The manuscript is well written and organized. The method is original and scientifically sound. I suggest publication after minor revision. My specific comments are
(1) The authors present the comparison of chamber pressures predicted by LEM and FEM. FEM simulation of Piezo inkjet device is not an easy and straightforward task, which involves fluid-structure interaction. The authors should provide more details about FEM simulation so that FEM results can be trusted.
(2) The authors mentioned that xi(t) in Line162 pg 5 as the pressure differences, whereas they indicated V (in eq. 9) line166 pg. 6 as the applied voltage. There is an inconsistence. What does eq. 9 represent?
(3) There are some typos in the paper, such as "For e.g.," line 48 pg 2, "A'"in eq. 2.
Author Response
Authors responses to 2nd reviewer comments
The authors thank the reviewer and the editor for providing constructive comments and suggestions to improve the scientific quality and readability of the manuscript. The author’s responses to the reviewer’s comments are below:
The authors present the comparison of chamber pressures predicted by LEM and FEM. FEM simulation of Piezo inkjet device is not an easy and straightforward task, which involves fluid-structure interaction. The authors should provide more details about FEM simulation so that FEM results can be trusted.
Author’s response: We thanked the reviewer for the valuable suggestion. More details about the FEM simulations are given below which will be added in the paper after subsection 2.2 of the paper.
FEM Simulations
The 3-dimensional model is simulated using numerical simulation software, COMSOL. Physics of fluid-structure interaction is used by considering the fluid as laminar and incompressible. As the design is symmetric, therefore, under laminar flow, symmetry is used as a boundary condition by considering the half of the 3-dimensional model to reduce the simulation time. As the pressure chamber is pressurized by deforming the piezo-membrane, a moving mesh is used by defining a mesh displacement on the wall at the top of the pressure chamber. The model is meshed by using tetrahedral, triangular and swept meshes. Structured mesh is used for the fluid in the restrictors, nozzle and pressure chamber. The meshed model is shown in figure 4.
The fluid flow in the micro-channels is described by the Continuity and Navier-Stokes equations [6]. For incompressible fluid, the continuity equation is given as
|
|
, |
(24) |
where is the divergence and is the fluid velocity. The Navier-Stokes momentum conservation equation of the motion for incompressible fluid is expressed as
|
|
, |
(25) |
where, is the density of the fluid, is the viscosity of the fluid, is the pressure, and is the unit diagonal matrix. Finally, the piezo-membrane is fixed from the around sides in the physics of solid mechanics. The physics of electrostatics is also used in FEM simulation by applying a voltage waveform (shown in figure 5) to the lower electrode of the piezo-membrane and considering the top electrode as ground. The study of time dependent is used to simulate the model by considering the time ranging from 0 to 60 with a time step of 0.1. The predicted pressure at the pressure chamber and velocity at the nozzle inlet have been obtained and compared with the LEM results as shown in figure 6
Figure 4. Meshed 3D model of half of the proposed recirculating inkjet printhead
References:
[6] Wei, H.; Xiao, X.; Yin, Z.; Yi, M.; Zou, H. A waveform design method for high DPI piezoelectric inkjet print-head based on numerical simulation. Microsyst. Technol., 2017, 23, 5365–5373.
The authors mentioned that xi(t) in Line162 pg 5 as the pressure differences, whereas they indicated V (in eq. 9) line166 pg. 6 as the applied voltage. There is an inconsistence. What does eq. 9 represent?
Author’s response: We thanked the reviewer for asking the practical question. In the fluid-electric analogy of the two-port lumped element modeling, the pressure (P) is considered as voltage and the volume flow rate (Q) as a current. Using equation (8) of the paper (, here Q is the volumetric flow rate which is considered as a current (I). So here, we use nodal analysis. The leaving currents from the node will be deducted from the entering currents to the node as shown in the figure below. The entering current to the node ( is the current across Re and the leaving current from the node is. That is the reason of using applied voltage in equation 9 of the paper.
There was another mistake in equation 9. Instead of (current source), we have written (voltage source), so we have also corrected that to.
There are some typos in the paper, such as "For e.g.," line 48 pg 2, "A'"in eq. 2.
Author’s response: We thanked the reviewer for pointing the typing mistake. The typo of “For e.g.,” is removed from the paper. The "A'" is not A-dash. It is actually a comma (,) after the eq. 2.
